20 GB in 10 minutes: a case for linking major biodiversity databases using an open socio-technical infrastructure and a pragmatic, cross-institutional collaboration

Thessen Anne E. annethessen@gmail.com 1 5
Poelen Jorrit H. 2
Collins Matthew 3
Hammock Jen 4
1 Ronin Institute for Independent Scholarship , Montclair , NJ , USA
2 Independent consultant , Oakland , CA , USA
3 University of Florida , Gainsville , FL , USA
4 National Museum of Natural History , Washington, DC , USA
5 Oregon State University , Corvallis , OR , USA
Frigeri Alessandro
Electronic publication date: 2018 Sep 17
Publication date: 2018
Volume: 4
Electronic Location ID: e164
Received 2018 May 17; Accepted 2018 Aug 29
Copyright: ©2018 Thessen et al.
Copyright year: 2018
Copyright holder: Thessen et al.
License: This is an open access article distributed under the terms of the Creative Commons Attribution License, which permits unrestricted use, distribution, reproduction and adaptation in any medium and for any purpose provided that it is properly attributed. For attribution, the original author(s), title, publication source (PeerJ Computer Science) and either DOI or URL of the article must be cited.
License URL: https://creativecommons.org/licenses/by/4.0/

Keywords: Biodiversity, Collaboration, Identifiers, Wikidata, Graph, Linking

Funding: NSF award 1547229 Funding was provided by David Rubenstein and the Encyclopedia of Life and by iDigBio, NSF award 1547229. The funders had no role in study design, data collection and analysis, decision to publish, or preparation of the manuscript.

==============================
Biodiversity information is made available through numerous databases that each have their own data models, web services, and data types. Combining data across databases leads to new insights, but is not easy because each database uses its own system of identifiers. In the absence of stable and interoperable identifiers, databases are often linked using taxonomic names. This labor intensive, error prone, and lengthy process relies on accessible versions of nomenclatural authorities and fuzzy-matching algorithms. To approach the challenge of linking diverse data, more than technology is needed. New social collaborations like the Global Unified Open Data Architecture (GUODA) that combines skills from diverse groups of computer engineers from iDigBio, server resources from the Advanced Computing and Information Systems (ACIS) Lab, global-scale data presentation from EOL, and independent developers and researchers are what is needed to make concrete progress on finding relationships between biodiversity datasets. This paper will discuss a technical solution developed by the GUODA collaboration for faster linking across databases with a use case linking Wikidata and the Global Biotic Interactions database (GloBI). The GUODA infrastructure is a 12-node, high performance computing cluster made up of about 192 threads with 12 TB of storage and 288 GB memory. Using GUODA, 20 GB of compressed JSON from Wikidata was processed and linked to GloBI in about 10–11 min. Instead of comparing name strings or relying on a single identifier, Wikidata and GloBI were linked by comparing graphs of biodiversity identifiers external to each system. This method resulted in adding 119,957 Wikidata links in GloBI, an increase of 13.7% of all outgoing name links in GloBI. Wikidata and GloBI were compared to Open Tree of Life Reference Taxonomy to examine consistency and coverage. The process of parsing Wikidata, Open Tree of Life Reference Taxonomy and GloBI archives and calculating consistency metrics was done in minutes on the GUODA platform. As a model collaboration, GUODA has the potential to revolutionize biodiversity science by bringing diverse technically minded people together with high performance computing resources that are accessible from a laptop or desktop. However, participating in such a collaboration still requires basic programming skills.

Introduction

Biodiversity databases provide global access to information about species via the Web. These databases contain information as varied as observation records, text descriptions, images, maps, genetic sequences, phylogenetic trees, and trait data (Table 1). All of these data become much more useful if they can be linked. Many biodiversity databases share information with each other (Bingham et al., 2017), but creating the links can be very difficult for several reasons including the size of the databases, the heterogeneous nature of the data, and the heterogeneous nature of the identifiers used by the different resources (Page, 2008).

The more popular methods for linking biodiversity databases include taxonomic names, LSID (Life Sciences Identifier), and DOI (Digital Object Identifier). The Encyclopedia of Life uses taxonomic names to automatically aggregate data from hundreds of providers (Parr et al., 2014). BioNames links data using LSID, DOI, handles, bibliographic citations, and taxonomic names (Page, 2013). The iPhylo LinkOut service mapped identifiers used by the NCBI taxonomy database (which provides the taxonomic backbone for GenBank) to Wikipedia pages using taxonomic names, including synonyms (Page, 2011). TBMap provides links from TreeBase across several taxonomic databases, such as ITIS and NCBI (Page, 2007). This mapping was also achieved using taxonomic names, but in some cases GenBank Accession numbers and museum specimen codes were available for supplement. The use of taxonomic names to aggregate data can lead to errors and requires significant a priori knowledge either in the form of curators or an authoritative nomenclature.

Many databases expose their own internal identifiers, such as the WoRMS Aphia ID, so others can link their data to those resources within their own systems, often by providing a URL. Databases like WoRMS provide web services that allow users to look up an identifier for a taxon in question, one at a time. While this makes linking easier, it is still difficult to scale across all databases. For example, a list of all the taxon identifiers in EOL is 300 MB compressed. No system of identifiers is universal across biodiversity databases and none of them are easy to implement at scale.

While the data would be much more useful if linked, there is a lack of tools for linking data across databases at scale. Most mappings are done at great expense and then are made available as a separate file or incorporated into the resources themselves. LinkOut, BioNames, GBIF, and EOL take more than a day to link across their entire body of aggregated content. This paper discusses links made between GloBI and Wikidata (WD) in 10 min using GUODA, a high performance computing system available for analysis of large biodiversity data sets.

Table 1 Selected biodiversity databases and their size.

Database	Data Quantity (Jan 2018)	Size (compressed)	
GBIF	964,547,793 occurrence records	139 GB	
Catalogue of Life/ITIS	1.7 million taxa	2.9 GB	
GloBI	3,363,528 interactions	206 MB	
iDigBio	106,922,498 specimen records	35.5 GB	
GenBank	206,293,625 sequences	3 TB	
Biodiversity Heritage Library	53,739,062 pages	2.7 GB	
WoRMS	243,323 marine species	71 MB	
OpenTree	2,722,024 taxa and 6,810 trees	189 MB	
EOL TraitBank	Over 11 million records	46 GB uncompressed	
EOL	7,705,748 data objects (May 2017)	10 TB uncompressed	
Wikidata	42,648,426 data items	20 GB	

Methods

Description of Resources

GUODA

Following an iDigBio hack-a-thon in June 2015, GUODA was created as a pragmatic way to compute over multiple large biodiversity databases in a mutually beneficial collaboration between iDigBio, EOL, Kew Garden, and independent developers. Catalyzed by various presentations at conferences, hardware provided by ACIS, 20+ meetings, and several prototypes (e.g., http://effechecka.org, https://gimmefreshdata.github.io), a general access biodiversity data integration and analysis environment was created. This environment, with the aggregated experience and perspectives of all the collaborators, was used to produce the results of this paper.

Housed at the ACIS Lab at the University of Florida, the GUODA infrastructure consists of 12 IBM HS22 blades each with 8 cores, 24 GB of memory, and 1 TB of storage each. This makes a total of 192 threads, 288 GB of memory and 12 TB of disk space available for processing jobs using Apache Spark (Fig. 1; Zaharia et al., 2016). The cluster is managed under Apache Mesos (Hindman et al., 2011) which is a distributed scheduling system for periodic jobs. For long running processes, such as web APIs or databases, the Marathon (https://github.com/mesosphere/marathon) framework is run within Mesos. Marathon facilitates running always-up services with monitoring, automatic deployment of code, re-scaling to multiple nodes, and other management features. Mesos is responsible for accepting requests to start Spark frameworks, processes which do the actual computation and may span multiple servers, and allocation of resources requested by the framework.

Figure 1 GUODA Infrastructure.

Data from biodiversity databases is loaded into GUODA as Parquet files (Storage). When a user working in a Jupyter Notebook (Front-end Server) triggers a job interactively or via GitHub and Jenkins, the data are analyzed using Apache Spark (Compute Cluster). This infrastructure allows a user working from a laptop or desktop to compute over multiple biodiversity databases at once. All logos are provided by the organizations they represent and are used with permission.

Hadoop HDFS (Shvachko et al., 2010) is installed outside of Mesos directly on all 12 nodes of the cluster and provides redundant parallel shared storage to all nodes as well as the Jupyter notebook (Kluyver et al., 2016) server that provides a programming interface to end users. Each node has 1 TB of local disk storage for a total of about 3.5 TB of usable storage space for data files in Apache Parquet format. Spark is aware of the placement of data on an HDFS cluster and will divide processing among nodes in a way that prefers to read and write data that is local to the node to minimize network traffic.

Wikidata

Wikidata (WD) is a free and open knowledge base that provides structured data for WikiMedia projects (http://www.wikidata.org; Vrandečić & Krötzsch, 2014). Similar to Wikipedia, anyone can read or edit the resource. Information, including links to other resources, can be added to Wikidata using bots and batch imports through their Data Import Hub (https://www.wikidata.org/wiki/Wikidata:Data_Import_Hub). Wikidata information about taxa can be conceptualized as a graph linking related taxa to each other and identifiers from other databases to the taxa they represent (Fig. 2). Every taxon in Wikidata is issued a Wikidata identifier. While a public Wikidata SPARQL endpoint and associated tools (Voß, 2016) exist, these APIs are not suitable for batch processing. For example, when attempting to retrieve all taxa using the public SPARQL endpoint, a query timeout error was reported. In addition, the APIs are expected to return different results over time, so reproducing results is difficult if not impossible. This is why we used a JSON archive to access Wikidata (Wikidata, 2018).

Figure 2 Frequency of Wikidata taxa linked to biodiversity databases.

This graph shows the proportion of the approximately 2.3 million Wikidata taxa with zero, one, two, etc. links to external biodiversity databases (NCBI, ITIS, GBIF, EOL, FishBase, Index Fungorum and iNaturalist). The majority of Wikidata taxa had at least two links. A little more than 15% of Wikidata taxa had no links to external biodiversity databases.

GloBI

GloBI is a database of biotic interactions recorded as Organism_1:has_relationship: Organism_2 (Poelen, Simons & Mungall, 2014) per individual interaction observation or claim. GloBI uses a combination of web APIs, taxon archives, and name correction/parsing methods in an attempt to link names from species interaction datasets to existing sources. Spatial, temporal, and taxonomic coverage in GloBI is sparse and unevenly distributed (see Eltonian shortfall, Hortal et al., 2015), with spatial concentrations in Europe and North America and taxonomically concentrated in Arthropods, Fungi, and Plants. Only 8% of taxa in ITIS are also in GloBI. A detailed technical description of the GloBI data model and services has been published elsewhere (Poelen, Simons & Mungall, 2014). GloBI maintains a graph of related taxa and their identifiers from different databases (Poelen, Simons & Mungall, 2014). GloBI does not introduce its own taxon IDs. Instead, it records how names were mapped from a source name into an external taxonomic database using a taxon graph (see https://globalbioticinteractions.org/references). We used GloBI Taxon Graph v0.4.2 (Poelen, 2018b). This taxon graph links names and identifiers hierarchically and across resources.

Open Tree of Life Reference Taxonomy

To assess taxonomic ID coverage, the taxa in Wikidata and GloBI were compared to Open Tree of Life Reference Taxonomy (OTT 3.0; http://files.opentreeoflife.org/ott/ott3.0/ott3.0.tgz; Rees & Cranston, 2017). OTT was built using an automated algorithm with informed choices to aggregate and link existing naming authorities into a reasonably comprehensive, artificial, taxonomy. OTT contains 4,385,000 external links for 3,594,550 taxa aggregated and linked over five authorities (i.e., GBIF, IF, SILVA, WoRMS, NCBI).

Linking Wikidata And GloBI

Both Wikidata and GloBI have taxon graphs that map to identifiers from external databases (e.g., NCBI, ITIS, GBIF, EOL, Index Fungorum (IF), Fishbase and WoRMS). A Wikidata dump was loaded into GUODA and processed to extract taxon items (about 2.3 million) and their links to NCBI, ITIS, GBIF, EOL, IF, Fishbase and WoRMS. This was the Wikidata taxon graph. This taxon graph was loaded into a lookup table where each row contained an NCBI, ITIS, GBIF, EOL, IF, Fishbase or WoRMS identifier and the corresponding Wikidata identifier. The GloBI taxon graph was already in a similarly formatted lookup table. The taxon graphs in GloBI and Wikidata were mapped to each other with a join of the NCBI, ITIS, GBIF, EOL, IF, Fishbase or WoRMS identifiers of the respective lookup tables (Fig. 3). So, for each external identifier that occurred in both Wikidata and GloBI, the corresponding Wikidata identifier inserted in the GloBI lookup table. For instance, consider Wikidata taxon item Q140 (https://www.wikidata.org/wiki/Q140 accessed on 30 March 2018; Panthera leo) points to ITIS:183803. With the matching algorithm used, GloBI now considers WD:Q140 to be linked to all taxon entries that are considered the same as, or synonymous to, ITIS:183803.

Figure 3 Mapping taxon graphs across resources.

Both GloBI and Wikidata contain hierarchical taxon graphs with each taxon having a “star” of external identifiers. The taxa are mapped across these resources by comparing the portion of the graph with the external identifiers between nodes. In this example, the names and identifiers match perfectly, so a relationship between Panthera leo in GloBI and Panthera leo in Wikidata is inferred.

This final joined graph was saved into HDFS as a Parquet file and linked entries were appended to GloBI Taxon Graph from v0.3.0 onward (Poelen, 2018c). In addition, the GloBI ingestion engine was updated to automatically perform the taxon graph matching for future updates. This linkage enabled lookups of diet items of lions by Wikidata identifier via https://www.globalbioticinteractions.org/?interactionType=eats&sourceTaxon=WD%3AQ140 and facilitates future integration of species interaction data with Wikidata.

Taxon graph overlap and consistency

OTT, Wikidata, and GloBI taxon graphs maintain links to GBIF, IF, NCBI and WoRMS identifiers (referred to as external identifiers). The taxon graphs are considered to (partially) overlap if individual taxon IDs from different graphs have at least one external identifier in common. In addition, a taxon graph is inconsistent if a taxon ID links to multiple external identifiers from the same identifier scheme. Similarly, overlapping taxon IDs are said to be inconsistent if they link to multiple external identifiers from the same identifier scheme. Where overlap is a measure for taxon graph similarity, consistency can be seen as a way to measure the relative quality of (overlapping) taxon graphs.

For instance, let’s say that OTT:1087695 is linked to NCBI:191633, WoRMS:156905, and GBIF:1449280. In addition, WD:Q7247420 (https://www.wikidata.org/wiki/Q7247420) points to WORMS:156905, GBIF:1449280, and NCBI:191633. This would mean that links of these OTT and WD IDs overlap and are consistent, because they do not point to different names in same naming schemes (Fig. 4). However, when considering the GloBI taxon “ID” “GLOBI:null@Procladius sp1 M_PL_014”, multiple links to external IDs were found (e.g., NCBI:1981571, NCBI:1981569, NCBI:1981572, NCBI:1981573, NCBI:1981574, NCBI:1981570). In this case, the GloBI taxon ID is inconsistent. The high number of external NCBI identifiers is due to the NCBI taxonomy containing many “provisional” taxa derived from environmental samples.

Figure 4 Inconsistent graph matching.

When overlapping taxon graphs include multiple name strings, the graph is inconsistent. In this example the Procladius genus is present in Wikidata (red), Open Tree (textured fill), and GloBI (blue). The Wikidata, OTT, and GloBI taxon graphs overlap on the NCBI and the GBIF identifiers (purple and textured fill). The WoRMS identifier overlaps the OTT and Wikidata taxon graph (red and textured fill). The Procladius graph in GloBI includes NCBI identifiers with a different name string, Procladius (Holotanypus), which indicates inconsistent usage.

Data access

All of the input data sets can be found at: https://doi.org/10.5281/zenodo.755513 (GloBI Taxon Graph), http://files.opentreeoflife.org/ott/ott3.0/ott3.0.tgz (Open Tree of Life Taxonomy) http://doi.org/10.5281/zenodo.1211767 (Wikidata).

A selection of intermediary and result datasets are available online (Poelen, 2018d; Poelen, 2018a).

All of the scripts used to make the statements in the results can be found here (https://github.com/bio-guoda/guoda-datasets/tree/master/wikidata) with instructions on how to duplicate the analysis.

Results

After 10 min of processing, GloBI was linked to Wikidata using pre-existing identifier mappings. The Wikidata dump was 20 GB of compressed JSON with 40–50 million data items. It took about 10 min for GUODA to extract taxa (about 2.3 million) and their links in Wikidata and then less than one minute to map the Wikidata taxon graph to the GloBI taxon graph. The 119,957 WikiData links that were added to GloBI increased its outgoing name links by 13.7% (Poelen, 2018d). Eighty-seven percent (86.7%) of the external identifiers in Wikidata overlap with the external identifiers in OTT (Fig. 5). Eighty-six percent (86.1%) of the external identifiers in GloBI overlap with the external identifiers in OTT (Fig. 5). Wikidata provided mappings for 65.2% of the external identifiers in GloBI (Fig. 5). Out of the 77,000 external identifiers that occurred only in OTT and GloBI, only 56 were inconsistent (https://github.com/bio-guoda/guoda-datasets/blob/master/wikidata/inconsistentNameIdsGloBI_OTT.tsv). These 56 links pointed to seven OTT “taxa”. No inconsistent links were found between WD and GloBI. Out of the 38,000 links only found in GloBI, 9,000 were inconsistent (https://github.com/bio-guoda/guoda-datasets/blob/master/wikidata/inconsistentNameIdsGloBIOnly.tsv). The OTT, Wikidata, and GloBI identifier graphs related to this coverage analysis is a 74 MB compressed tab-separated-values file consisting of about 12 million identifier mapping records (see https://zenodo.org/record/1213477/files/links-globi-wd-ott.tsv.gz). The resulting Wikidata taxon objects were merged into GloBI’s Taxon Graph (Poelen, 2018d).

Figure 5 Identifier overlap between Wikidata (WD), OTT, and GloBI.

This Venn Diagram shows the number of overlapping external identifiers that can be found in one of three databases. Only 207,958 external IDs can be found in all three. These consisted of 22,637 WoRMS links, 71,980 NCBI links, 103,300 GBIF links and 10,040 IF links. Over two million IDs are only known to one of the three databases. OTT contains more than half of the external IDs in Wikidata and in GloBI, but neither contain half of the external IDs in OTT. Mapping Wikidata to GloBI matched 65.2% of the external IDs in GloBI.

In order for a mapping to be considered consistent, there can only be one identifier per resource included in each local graph. Thus, after removing the inconsistent identifiers, the external ID overlap can be interpreted as an estimate of the number of shared taxon names between two databases (Table 2). This cannot be interpreted as total taxa in each resource.

Table 2 Absolute and relative link counts from OTT, WD, and GloBI compared to WoRMS, GBIF, Index Fungorum (IF), and NCBI.

	WoRMS	GBIF	IF	NCBI	Combined	
OTT	327,929 (100%)*	2,451,566 (100%)	276,262 (100%)	1,355,207 (100%)	4,410,964 (100%)	
WD	288,110 (88%)	1,779,789 (73%)	76,497 (28%)	410,092 (30%)	2,554,488 (58%)	
GloBI	68,565 (21%)	315,173 (13%)	33,400 (12%)	704,361 (52%)	1,121,499 (25%)	
Notes.

* Overlap between each resource and OTT is set at 100%. The other percentages give a relative estimate of size and scale and should not be interpreted as overlapping IDs.

Discussion

GUODA is a high performance computing resource for biodiversity science that provides scalable solutions for working with large data sets in a collaborative, online environment. The 10 min processing time for 20 GB of compressed JSON is far faster than any current mapping method used in biodiversity; however, it does benefit from the mapping already completed inside Wikidata. For example, the Wikidata entry for Panthera leo (https://www.wikidata.org/wiki/Q140) has 25 links to external databases, not all of them biodiversity-related. This linking may be based on matching name strings. Other efforts using name-string-matching to link biodiversity databases take much longer to map resources together. For instance, EOL takes more than a day to map the content it receives from providers to a unified classification (J Rice, pers. comm., 2018). Similarly, the taxon matching in BioNames and LinkOut took days to complete (R Page, pers. comm., 2018). Projects like OTT, Wikidata, and GloBI that keep identifier-based taxonomic graphs make it easier to link databases at scale.

Despite the notoriously poor nature of taxon names as identifiers, they are still commonly used to link biodiversity data. A much-discussed solution has been the use of universal, unique, persistent, resolvable identifiers across the biodiversity data landscape, but the social barrier to a universal identifier system has, thus far, proven insurmountable (Nimis, 2001; Hardisty, Roberts & The Biodiversity Informatics Community, 2013). Rather than rely on name strings or a universal identifier system, this method uses the graph of identifiers to map taxa across two databases. This identifier-based method has the potential to be faster and easier than name-string matching without some of the social difficulties of a single identifier system.

Most biodiversity databases and nomenclatural authorities expose their data in idiosyncratic ways that are not suitable for batch processing. If data sources published their taxon identifier graph as a lookup table (as described in this paper) integrating across databases would be much easier (Fig. 6). Now, users have to learn a unique format for every data source. These lookup tables have the advantage of being easy to version and integrate.

Figure 6 Example look up table.

This figure is an excerpt from the GloBI look up table. The providedTaxonId and the providedTaxonName come from the taxon graph external to GloBI. The resolvedTaxonId and the resolvedTaxonName are the names and identifiers that are already mapped within GloBI. Each row represents a mapping from a taxon in an external source (Pluvialis obscura) to an identifier from a source already in GloBI, which does not mint its own identifiers.

In addition to fast linking of biodiversity databases, comparison of identifier graphs may be a scalable way to find inconsistencies, especially when multiple biodiversity databases/identifiers are included. By linking GloBI to OTT and WD, inconsistent names or false positive name matches were detected by considering the (lack of) overlap of GloBI names with OTT and WD external identifier schemes. These inconsistencies might be introduced by a dataset or a name resolution method that produces ambiguous results. In addition, inconsistencies can indicate a disputed/outdated name like “GLOBI:null@Senecio pectinatus” which maps to GBIF:8317096 and GBIF:8414746. This would be considered an inconsistent mapping and suggests that Senecio pectinatus is an outdated name. A related method using a variation of the PageRank algorithm (Page et al., 1999; Brin & Page, 1998) to identify the most legitimate taxonomic name to apply to a fossilized specimen (Huber & Klump, 2009) gives further legitimacy to this concept. Combining the speediness with the promise of scalability, a near-real-time name consistency check can be implemented to detect inconsistencies across various systems in the biodiversity data-ecosystem introduced by integration bugs, taxonomy updates or differences of interpretation.

GUODA has been available since 2015 and contains data dumps from GBIF, EOL TraitBank, iNaturalist, iDigBio, and BHL which are all accessible via a Jupyter notebook, web services, or Apache Spark shell on the command line. Despite its computing power and successful demonstrations at major conferences, GUODA has not been used to its full potential. The barrier of learning new programming and computing paradigms as well as developing an understanding of large dataset work flows seems to be a barrier to many in the biodiversity community. Despite this, GUODA is being used in several capacities. The Effechecka application generates taxonomic checklists using a web interface that allows a user to draw a polygon on a map and returns a deduplicated list of taxa aggregated from observation data held in GBIF, iNaturalist, etc. The EOL Freshdata project uses it to enable the detection of new occurrence records given geospatial and taxonomic and data source constraints and notifies interested users via email. Several workshops have used it to teach Spark programming skills to students at the University of Florida.

Future work on the GUODA infrastructure includes training and evaluating neural network models on image data, containerization of the GUODA components to allow the system to be run in additional data centers, and refinement of the end-user interface to integrate programming, source code, and publication to make research more reproducible. GUODA’s most impactful contribution has likely been the availability of readily formatted biodiversity data and new data sets will continue to be added to the collaboration platform, enabling domain experts and technical experts to answer new questions in the future.

The bottlenecks in processing for Hadoop File System and Apache Spark are the number of CPUs, amount of memory, and available storage space allocated to the computer cluster. Both HDFS and Spark are designed to scale horizontally by adding commodity servers (aka nodes) to increase the processing power, working memory, and storage space. Thus, this problem is immediately solvable. Internet bandwidth to transfer the data archives from Open Tree of Life, Wikidata, and GloBI does not scale and is not something that can be addressed solely within our research group. At the moment, it takes longer to download the Wikidata resource than it does to run the linking process discussed in this manuscript. Socio-technical bottlenecks include resource-limitation and user education. Increased usage and operational support is expected to positively impact processing performance by encouraging pro-active bug fixing and infrastructure maintenance. In addition, while the technical complexity of operating and using a compute cluster have been dramatically reduced since the introduction of Hadoop in 2006, some re-education may be needed to effectively use these powerful data tools (e.g., jupyter notebooks, HDFS, scala).

GUODA, and hosted data analytics infrastructure in general, has the potential to drastically improve biodiversity science by making multiple biodiversity databases accessible to scientists for analysis on their laptop or desktop. Users still need to have some programming skills, which have now become an essential skill in biodiversity science.

Conclusions

Sharing information between biodiversity databases can be difficult because of the amount and heterogeneity of the data and the identifiers. Most mappings are done using taxonomic name strings at great expense. We were able to map Wikidata to GloBI in 10 min using identifier graphs and GUODA, a high performance computing infrastructure developed through collaboration between diverse players. The mapping increased GloBI’s outgoing name links by 13.7%. This method of mapping across databases using identifier graphs is faster than comparing name strings and can help find inconsistencies that point to a disputed or outdated name. GUODA, and systems like it, have the potential to revolutionize biodiversity science by bringing diverse technically minded people together with high performance computing resources that are accessible from a laptop or desktop.

The authors would like to acknowledge support and resources provided by the ACIS lab. The authors would like to acknowledge José A.B. Fortes for providing infrastructure and creating room for collaboration. The authors would like to thank the two reviewers for their insightful comments that greatly improved the manuscript. The Encyclopedia of Life and iDigBio helped establish an informal yet pragmatic cross-institutional collaboration.

Additional Information and Declarations

Competing Interests

Author Contributions

Data Availability

The authors declare there are no competing interests.

Anne E. Thessen analyzed the data, prepared figures and/or tables, authored or reviewed drafts of the paper, approved the final draft.

Jorrit H. Poelen conceived and designed the experiments, performed the experiments, analyzed the data, contributed reagents/materials/analysis tools, prepared figures and/or tables, performed the computation work, authored or reviewed drafts of the paper.

Matthew Collins contributed reagents/materials/analysis tools, performed the computation work, authored or reviewed drafts of the paper.

Jen Hammock authored or reviewed drafts of the paper, provided collaborative space and leadership.

The following information was supplied regarding data availability:

Poelen, Jorrit H. (2018). Global Biotic Interactions: Taxon Graph (Version 0.3.5) [Data set]. Zenodo. http://doi.org/10.5281/zenodo.1313243.

WikiData. (2018). Wikidata dump 2017-12-27 [Data set]. Zenodo. http://doi.org/10.5281/zenodo.1211767

Poelen, Jorrit. (2018). 20 GB in 10 min: Data linking across major biodiversity databases: Data supplements (Version 0.1) [Data set]. Zenodo. http://doi.org/10.5281/zenodo.1213477

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
