# Peer review of "GB in 10 minutes: a case for linking major biodiversity databases using an open socio-technical infrastructure and a pragmatic, cross-institutional collaboration"

_PeerJ Computer Science, doi:10.7717/peerj-cs.164_

## Round 0.1 · original submission · Minor Revisions

Dear Anne,

This paper touches the problem of the interoperability of databases, which is very relevant in most up-to-date scientific research activities.

Although this applies to the field biodiversity I see the potential for the application of the approach presented in this paper also in other disciplines.

Therefore I'm strongly towards the publication of this paper after taking into consideration the minor revisions suggested by the two reviewers, which will surely improve the already good research paper you have submitted to PeerJ.

Best Regards,

Alessandro Frigeri

·

Basic reporting

no comment

Experimental design

no comment

Validity of the findings

no comment

Additional comments

This is a very exciting paper that outlines a methodology for aligning large biodiversity datasets in a very short amount of time (certainly, less time than it takes to download some of the data). The approach is a significant advance in an area that has struggled to map data across multiple data providers.

I have a few minor quibbles, mostly about how the work is presented. These are listed below, in no particular order.

The phrase "taxon graph" was a little unclear. If I'm correct, it's simply a "star tree" with a node in source database in the centre, and each external identifier that it is linked to is represented by a node connected to that central node. To treat this as a graph seems overly elaborate, why not simply treat it as a set of external identifiers?

The description of overlap and consistency (line 169 onwards) might benefit from a diagram of some examples, rather than require the reader to keep a set of identifiers in their head. It might also be useful to explain why "GLOBI:null@Procladius sp1 M_PL_014" had so many mappings.

The performance is impressive, but the comparison to other efforts (e.g., EOL, BioNames, line 230) suffers somewhat because it relies on anecdotal recollections of how much effort was expended on semi-automated mapping. However, there's not much which can be done about this.

The authors recommend (line 245) that owners of taxonomic databases provide lookup tables to help data integration. The description of a look up table is verbal, and so open to some ambiguity. A diagram of an example would make it very clear what format is expected (I'm reminded of how explicit the "Tidy data" paper is in this regard, see http://vita.had.co.nz/papers/tidy-data.html).

The Venn diagram in Fig 3 is not to scale, which is somewhat misleading. How about using something like a cluster map, see http://iphylo.blogspot.com/2012/06/visualising-differences-between.html I've attached a very crude version to show what I mean. Such a diagram makes it clear to what extent the various sources overlap, without the distortion of the Venn diagram (the circles could be scaled to the log of the number of taxa in each category, for example).

Looking at the github repository for this project (https://github.com/bio-guoda/guoda-datasets/blob/master/wikidata/README.md) I'm invited to go to https://guoda.bio/ to learn more. There's a problem with the HTTPS security certificate for this site, which triggers all sorts of warnings in various web browsers.

DOI and LSID are acronyms and should be capitalised, and perhaps spelt out the first time they are mentioned.

·

Basic reporting

The manuscript is well written and well structured. It is easy to read and to follow the authors' reasoning. The literature references provide sufficient context.

Figure 1 is a bit oversimplified. In addition to product logos, it would be helpful to learn about the functional elements of the architecture.

Experimental design

The use of Wikidata as a source of taxonomic information is highly novel and well executed.

As part of the discussion (line 241-242), the authors mention the social barriers that have prevented the biodiversity community from introducing identifiers for taxa. A paper by P. Nimis (2001) illustrates these barriers quite well and might be included as a reference.
Nimis, P. L. (2001). A tale from Bioutopia - Could a change of nomenclature bring peace to biology’s warring tribes? Nature, 413(6851), 21. https://doi.org/10.1038/35092637

Further on in the discussion (lines 250 f), the authors note that a detection of inconsistencies in the identification of species across systems might also be implemented. It would be of interest to learn why this has not been implemented, even if it was just due to a lack of time or resources. A possible approach to dealing with taxonomic inconsistencies has been demonstrated by Huber & Klump (2009), showing that the validity of taxonomic names can be estimated by graph analysis.
Huber, R., & Klump, J. (2009). Charting taxonomic knowledge through ontologies and ranking algorithms. Computers & Geoscience, 35(4), 862–868. https://doi.org/10.1016/j.cageo.2008.02.016

Validity of the findings

In their paper, the authors stress several times that their solution is very fast. The manuscript would be more informative in this respect of the authors could supply the reader with more information, if available, about where the bottlenecks in processing are and how the system scales.

Additional comments

In general, I enjoyed reading this manuscript. The use of Wikidata and JSON outlines a new and original approach to biodiversity data that will be able to bridge the gaps between existing biodiversity information systems at scale.

---

## Round 0.2 · accepted · Accept

Dear Anne, I'm glad to see the good use of reviewers' comments, resulting in an improved version of your already good article.

I do believe that this will be an important reference for quick data access from different sources. The problem is relevant to other disciplines as well and I'm glad to see works being made in this direction.

I'm completely favorable to the publication of the article in PeerJ Computer Science.